# The phage-encoded PIT4 protein affects multiple two-component systems of *Pseudomonas aeruginosa*

Kaat Schroven,[1] Leena Putzeys,[1] Alison Kerremans,[1] Pieter-Jan Ceyssens,[1] Marta Vallino,[2] Jan Paeshuyse,[3] Farhana Haque,[4] Ahmed Yusuf,[4] Matthias D. Koch,[4] Rob Lavigne[1]

**ABSTRACT** Two-component systems (TCSs) control a large proportion of virulence factors in *Pseudomonas aeruginosa*. Yet, investigations on inhibitors of regulatory pathways of TCSs remain scarce, despite their potential in anti-virulence strategies. This work elucidates the molecular mechanism of PIT4, a protein encoded by the lytic *P. aeruginosa* phage LSL4. Single-copy expression of this early phage gene inhibits bacterial motility, in particular twitching motility, and reduces the virulence of *P. aeruginosa* in HeLa cells. Differential gene expression and a yeast two-hybrid screen showed that PIT4 interacts with components of different two-component systems. In one-on-one interaction assays, it was confirmed that PIT4 specifically interacts with the histidine kinase domains of FleS, PilS, and PA2882. This identified phage mechanism therefore demonstrates the ability of phage proteins to simultaneously target and impact multiple pathways and hints toward a biological function as an infection-exclusion mechanism. This work highlights the potential of previously unknown phage proteins in virulence regulation of multidrug resistant pathogens that could in future be exploited for anti-virulence strategies and biotechnological applications.

**IMPORTANCE** More and more *Pseudomonas aeruginosa* isolates have become resistant to antibiotics like carbapenem. As a consequence, *P. aeruginosa* ranks in the top three of pathogens for which the development of novel antibiotics is the most crucial. The pathogen causes both acute and chronic infections, especially in patients who are the most vulnerable. Therefore, efforts are urgently needed to develop alternative therapies. One path explored in this article is the use of bacteriophages and, more specifically, phage-derived proteins. In this study, a phage-derived protein was studied that impacts key virulence factors of the pathogen via interaction with multiple histidine kinases of TCSs. The fundamental insights gained for this protein can therefore serve as inspiration for the development of an anti-virulence compound that targets the bacterial TCS.

**KEYWORDS** bacteriophages, TCS, HK, FleS, PilS, T4P, motility, *Pseudomonas aeruginosa*, virulence

*P*seudomonas aeruginosa is a Gram-negative, opportunistic human pathogen and is endowed with an extensive set of virulence factors, enabling it to cause both acute and chronic infections (1). For instance, lipopolysaccharides (LPS) and motility moieties contribute to cell colonization and secretion systems, while effector molecules can cause tissue damage and interfere with host factors. Moreover, biofilms protect cells from antibiotics and other defense mechanisms (2). The repertoire of virulence factors encoded by *P. aeruginosa* is largely regulated by two-component systems (TCSs), since more than 50% of the bacterial TCSs are involved in virulence regulation (1). TCSs consist of a sensor kinase [histidine kinase (HK)] and a response regulator (RR), which is activated by the stimulated kinase upon an environmental stimulus. In turn, this will

Address correspondence to Rob Lavigne, rob.lavigne@kuleuven.be.

The authors declare no conflict of interest.

See the funding table on p. 15.

elicit a response inside the cell. Classically, the HKs are composed of an N-terminal signal sensing domain and a C-terminal transmitter domain, harboring a conserved histidine residue. The N-terminal domain is highly variable and often contains multiple hydrophobic regions for cell membrane imbedding. The RRs all have a conserved N-terminal receiver domain and a highly variable response domain (3).

For instance, the Gac/Rsm cascade is responsible for the transition of acute to chronic infection by controlling the expression of two non-coding RNAs (4). An unknown environmental cue activates *gacS*, which eventually leads to the upregulation of genes involved in biofilm formation, the type VI secretion system and results in a downregulation of the type III secretion system and flagellar motility (5). By contrast, flagellum-dependent motility is controlled by FleS/FleR, which is responsible for the regulation of over 20 flagellum biosynthesis-associated genes, which in turn drives the switch between acute and chronic infection. Remarkably, unlike other HKs, FleS lacks the typical transmembrane component and is therefore present in the cytoplasm (6). Furthermore, it was reported that only FleR is essential for the regulation of swimming motility, as the motility of a PAO1Δ*fleS* mutant was only partly affected (7). Interestingly, twitching motility is controlled by a number of TCSs. On the one hand, the FimS/AlgR system controls the operon encoding the minor pilins and PilY1 adhesin, while ChpA/PilG are crucial for the regulation of pilus extension and retraction (8–10). On the other hand, PilS/PilR is involved in the regulation of *pilA* expression and is composed of the sensor PilS, with six-transmembrane segments that sense PilA. The regulator, PilR, binds to $\sigma^{54}$ after activation and activates transcription of the *pilA* promoter (11, 12). Recently, it has been shown that the PilS/PilR system not only regulates the T4P but also controls the expression of genes with a role in iron uptake, flagellar assembly and function, including the *fleSR* regulon, and cell envelope synthesis (6).

Bacteriophages are known for their efficient takeover of the host cell and interact with key regulatory elements to shunt the bacterial cell metabolism toward progeny production (13). For example, *gp2* of *Pseudomonas* phage φKMV encodes for an inhibitor of the RNA polymerase, favoring the transcription of the viral genome (14). Furthermore, the phage-derived protein CP12 inhibits the Calvin cycle of cyanobacteria to redirect the host energy stocks (15). It has been recently reported that TCSs, and more particularly RetS-GacS/GacA, play a role in phage infection regulation by affecting the T4P-mediated absorption (16). Despite the importance of two-component regulators in bacteria, no phage protein has yet been reported that directly interacts with a component of a TCS. In this work, we describe a unknown protein derived from the strictly lytic *P. aeruginosa* phage LSL4, named PIT4, that interacts with three different TCSs, PilS, FleS, and PA2882, via the histidine kinase domain and consequently impacts among others the overall motility of the bacterial cell through these molecular interactions.

## RESULTS AND DISCUSSION

### Screening for unknown phage proteins impacting bacterial motility reveals PIT4

A systematic screen to identify gene products of lytic *P. aeruginosa* phages that impact motility of the bacterial host cell was performed on a previously described library of over 150 *P. aeruginosa* PAO1 strains, harboring a genomic integration of a gene encoding for an early phage protein (Fig. S1) (17). This assay revealed two distinct phage-derived proteins that significantly decrease bacterial twitching motility (Fig. 1; Student's *t*-test, $P < 0.0001$). Remarkably, twitching motility is completely abolished upon production of the protein gp67 of phage LSL4 (hereafter called PIT4: phage inhibitor of the T4P). The *pit4* gene encodes a 131 aa protein with no conserved domains or known homologs among the genomes of other *Nankokuvirus* members, of which LSL4 is a new isolate described here (OQ970155).

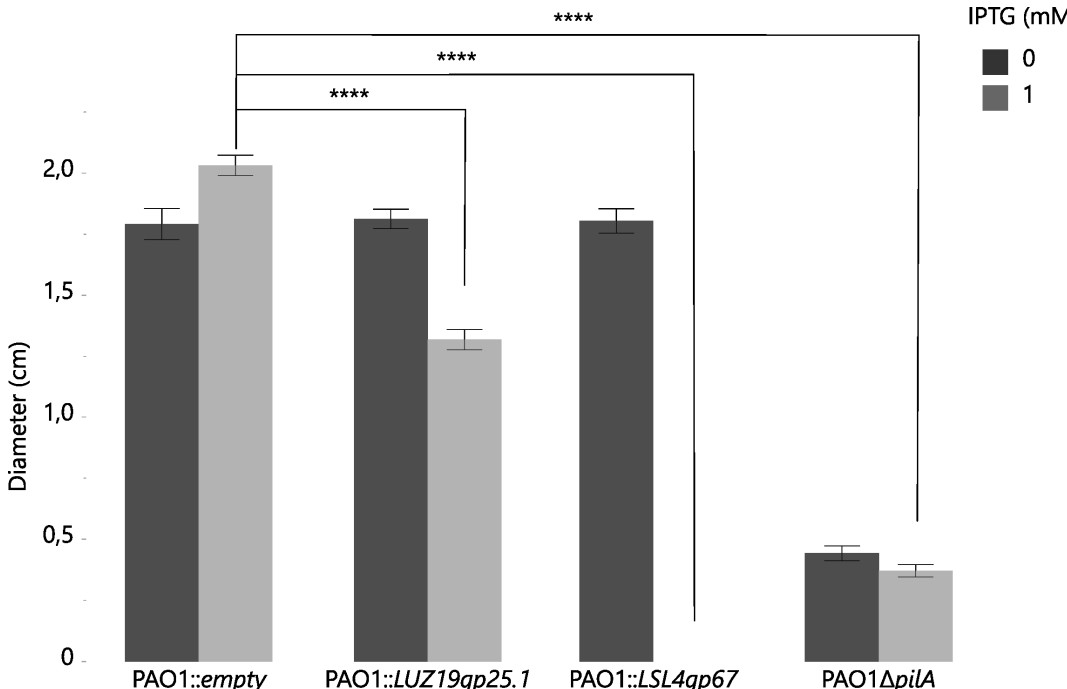

FIG 1 Expression of phage-derived genes impacting the bacterial twitching motility. The overexpression of two phage-derived genes in the PAO1 cell significantly reduces the bacterial twitching motility, when compared to the negative control (PAO1::empty, Student's *t*-test, ****$P < 0.0001$). Remarkably, production of LSL4gp67 (named PIT4, phage inhibitor of the T4P) resulted in a complete inhibition of twitching movement. As a negative control, the PAO1 transposon strain defective in PilA production was used.

## PIT4 impacts the infection of *P. aeruginosa* in HeLa cells, but not *in vivo*

To study the impact of PIT4 on the PAO1 virulence in HeLa cells and *in vivo*, we constitutively expressed the phage gene in *P. aeruginosa* PAO1 (Fig. 2). First, we observed significantly increased viability of the HeLa cells compared to the negative controls PAO1 and PAO1::empty, which was also confirmed via microscopy (Tucky's test, $P < 0.0001$; Fig. 2A; Fig. S2). Nevertheless, a higher virulence compared to the positive control (HeLa cells in growth medium) was observed. Next, the effect *in vivo* was tested using the *Galleria mellonella* infection model. The immune system of the invertebrate larvae shows similarities to the human innate immune system and therefore serves as a useful tool to assess the *in vivo* virulence of pathogens (18). The PAO1::*pit4* and wild-type PAO1 strains were injected into the last proleg of the larvae, and their viability was assessed after 24 hours. The injection of PAO1::*pit4* and PAO1::*empty* resulted in a 10% and 0% survival rate, respectively (Fig. 2B), and consequently, production of the phage protein in the *Pseudomonas* cell did not cause a significant change in survival rate of the larvae in this setup (log-rank test, $P < 0.34$).

The contradictory results that were obtained from the cell line and *in vivo* assay could be explained by the targeted virulence factor by PIT4. Based on the results gained in the *in vitro* motility assay, we assume that PIT4 interrupts the action of the T4P. It is known that T4P is crucial to establish an infection. Previously, the reduced capacity of *pilT* mutants to internalize, replicate in, and exit HeLa cells has been observed (19). This is consistent with the observation made here, as PAO1 cells with impaired T4P functionality are less virulent toward the cultured HeLa cells. By contrast, PIT4 does not rescue the larvae from the pathogenic effects of PAO1. It has been previously reported that both extracellular proteases of *P. aeruginosa* and the T3SS are of importance in the virulence of the pathogen toward *G. mellonella* (20, 21). The correlation between the motility structures and the virulence against *G. mellonella* has not been established, and one could argue that those structures might be of lesser importance for infection of this host.

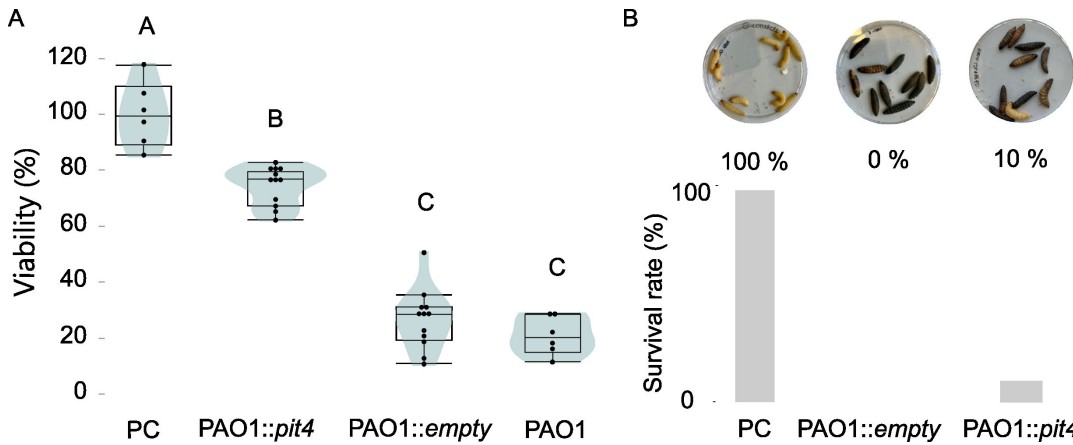

**FIG 2** PIT4 impacts the PAO1 virulence against HeLa cells but not in *G. mellonella*. (A) The virulence of different PAO1 strains in HeLa cells was assessed. HeLa cells in growth medium served as a positive control, while PAO1 and PAO1::empty were used as negative controls. Human cells exposed to strains producing PIT4 show a significantly higher viability when compared to the negative controls (Tukey's test, $P < 0.0001$). (B) The survival of *G. mellonella* larvae injected with different *P. aeruginosa* strains was determined. For the positive control, phosphate-buffered saline buffer was injected into the proleg. Expression of pit4 only resulted in 10% survival of the larvae and thereby did not significantly impact the PAO1 virulence compared to the negative control (PAO1::empty) (log-rank test, $P < 0.34$).

## RNA sequencing reveals an effect of PIT4 on different virulence-associated genes

To reveal the underlying mechanism of PIT4, we performed a differential gene expression experiment using RNA sequencing. For this, a PAO1::*empty* culture served as a control to analyze the genes that were differentially expressed in the PAO1::*pit4* strain. RNA from both strains was extracted in triplicate under induced and uninduced conditions. Principal component analysis indicated that PIT4 production results in an altered RNA expression profile in the bacterial cell (Fig. S3). In summary, 71 genes are differentially expressed (Log2FoldChange > |1.2| and $p_{adj} < 0.001$; Table S1) including 19 genes that are upregulated, whereas 52 genes are downregulated, compared to the induced condition of the control strain (Fig. 3). A first observation that can be made is the upregulation of the operon encoding for the subunits of the aa$_3$ cytochrome c oxidase [*coxAB* (*PA0106* and *PA0105*) and *coxIII* (*PA0108*)] (Log2FoldChange = 2.66, 2.57 and 2.32 and $p_{adj} < 0.001$, respectively). The upregulation of the terminal electron acceptor might be caused by nutrient starvation induced by the action of PIT4 (22). Interestingly, the aa$_3$ oxidase also can cause cell apoptosis by the generation of reactive oxygen species (23). Contrarily, the expression of *PA1183* or *dctA* is strongly downregulated and encodes for an H$^+$/dicarboxylate symporter protein (Log2fold = −4.74 and $p_{adj} < 0.001$). DctA proved to be essential for nitrogen fixation, and this C$_4$-dicarboxylate transport system is regulated by the TCS DctB/DctD (24, 25). The data interpretation provided further on focuses on motility-associated genes, considering the scope of this research.

The expression of multiple genes involved in the biosynthesis of the flagellum was downregulated. All the genes of the *flgBCDE* operon are significantly less expressed in the presence of PIT4 (Log2FoldChange <−1.8 and $p_{adj} < 0.001$) and encode structural parts of the flagellar apparatus. The TCS FleS/FleR positively regulates the operon, as well as the *fglFGHIJKL* and *fliK* operon, of which the latter gene was also differentially expressed in our data set (Log2FoldChange = −1.7 and $p_{adj} < 0.001$) (26). Furthermore, the flagellar motor protein encoded by *motY* as well as *gcbA*, which encodes a conserved diguanylate cyclase with a role in biofilm formation and flagellar motility, were also downregulated (Log2FoldChange = −2.0 and $p_{adj} < 0.001$ and Log2FoldChange = −1.7 and $p_{adj} < 0.001$, respectively). Interestingly, a recent publication reported that the transcription of *gcbA* is positively regulated by FleR (27). Lastly, the membrane-localized regulator MorA showed a significantly decreased expression (Log2FoldChange = −1.3

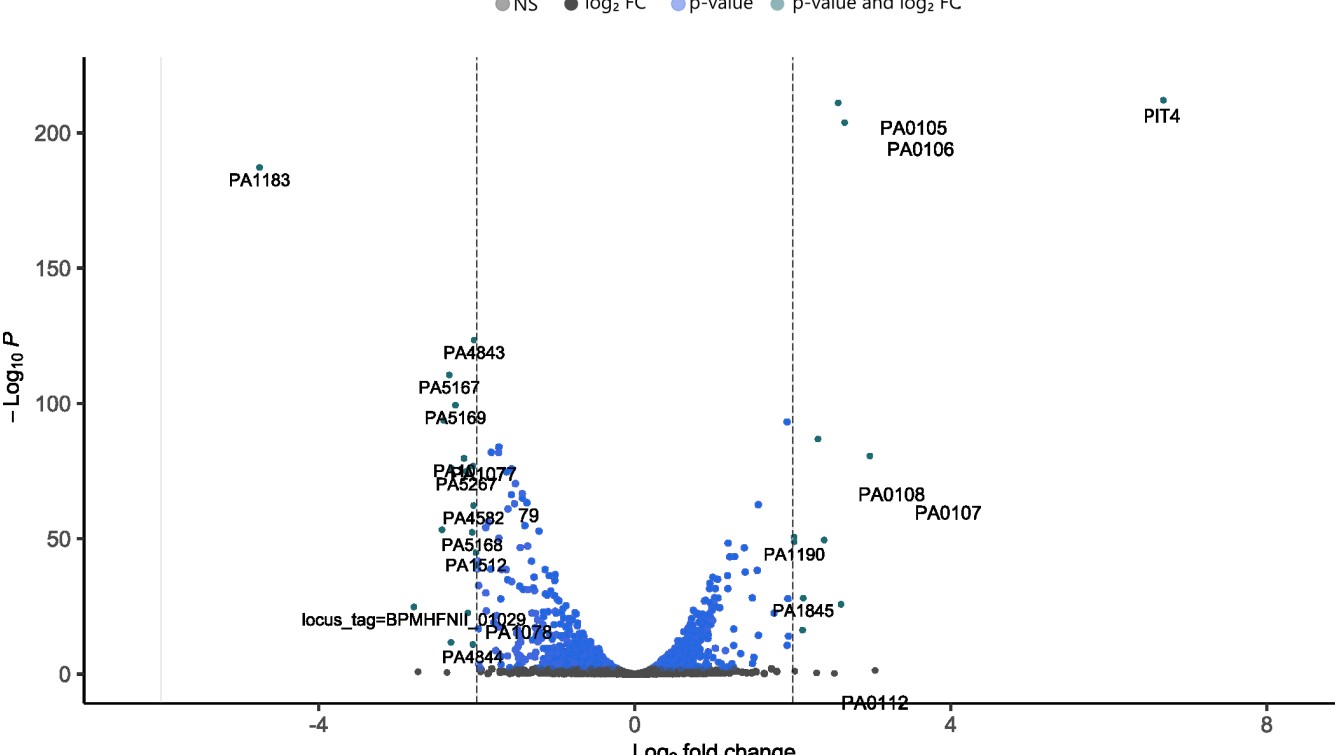

**FIG 3** Volcano plot of the differential gene expression assay. A differential gene expression experiment was performed to assess differentially expressed genes in the presence of PIT4 compared to a control strain. PIT4 production results in a change of expression for 71 genes (Log2FoldChange >|1.5| and $p_{adj}$ <0.001). In total, 52 genes are downregulated, whereas 19 genes are upregulated.

and $p_{adj}$ <0.001). This protein plays a role in the timing of flagellar development (28). Remarkably, only a few genes associated with the T4P were significantly downregulated. The two genes encoding for the major subunit of type IVa pili and the Tad pili, *pilA* and *flp*, were significantly less expressed (Log2FoldChange = −1.5 and $p_{adj}$ <0.001 and Log2FoldChange = −1.6 and $p_{adj}$ <0.001, respectively) (29). Moreover, downregulation of *amrZ* expression was seen, which gene product regulates twitching motility, more specifically the surface localization of the T4P, and alginate synthesis (Log2FoldChange = −1.6 and $p_{adj}$ <0.001) (30). Another downregulated operon upon *pit4* expression is *siaABCD*, which is associated with biofilm formation (Log2FoldChange <−1.38 and $p_{adj}$ <0.001). This operon encodes for a signaling network, in which the cyclic diguanylate cyclase SiaD plays a role in biofilm and aggregate formation by the production of c-di-GMP (31). Moreover, the transcription of *siaD* seems to be FleR-dependent (27). It should be noted that some of genes that are part of the type VI secretion system, genes fulfilling a role in iron acquisition, and a large number of hypothetical genes were downregulated as well (Table 1; Table S1)

## A yeast two-hybrid interaction assay indicates interaction of PIT4 with components of TCSs

To identify *P. aeruginosa* interaction proteins of PIT4, a yeast two-hybrid (Y2H) experiment was performed. PIT4 was used as bait, and the encoding gene was cloned into the pGBT9 plasmid. As prey, we made use of a previously constructed random genomic fragment library of *P. aeruginosa* PAO1 (13). Intriguingly, all seven potential interaction partners of PIT4 that were obtained played a functional role in different two-component systems of *P. aeruginosa* (Table 2). Two fragments of FleS were identified containing the histidine kinase moiety of the protein. Likewise, the histidine kinase domain of BfiS and the probable TCS components PA2177, PA2882, and PA1243 emerged from the

**TABLE 1** Differentially expressed genes under PIT4 producing conditions associated with motility[a]

| Locus tag | Gene name | Log2FoldChange | $p_{adj}$ |
|---|---|---|---|
| **Flagella-associated genes** | | | |
| PA1077 | flgB | 2.42 | 2.13E-94 |
| PA1081 | flgF | 2.32 | 9.26E-77 |
| PA1079 | flgD | 2.16 | 1.96E-80 |
| PA3526 | motY | 2.05 | 1.43E-77 |
| PA4843 | gcbA | 2.03 | 3.82E-124 |
| PA1078 | flgC | 2.01 | 7.93E-41 |
| PA1080 | flgE | 1.82 | 1.17E-82 |
| PA1441 | fliK | 1.72 | 1.30E-84 |
| PA4601 | morA | 1.28 | 1.48E-36 |
| **T4P-associated genes** | | | |
| PA3385 | amrZ | 1.60 | 9.72E-62 |
| PA4525 | pilA | 1.53 | 2.25E-88 |
| PA4306 | flp | 1.46 | 3.48E-33 |
| **Biofilm-related genes** | | | |
| PA0172 | siaA | 2.11 | 2.90E-23 |
| PA0169 | siaD | 1.47 | 8.51E-12 |
| PA0171 | siaB | 1.45 | 2.02E-12 |
| PA0170 | siaC | 1.38 | 5.08E-06 |

[a]Log2FoldChange >|1.2| and $p_{adj}$ <0.001.

interaction assay. Based on the Y2H results, we can conclude that the interaction partner of PIT4 is most likely a component of a TCS and more specifically a sensor kinase. It should be noted that most histidine kinases contain a transmembrane component (32). Therefore, we need to keep in mind that the folding of some protein fragments might be affected in this experiment.

## A broad interaction of PIT4 with several bacterial histidine kinase domains

To confirm the obtained results from the yeast two-hybrid experiment, a bacterial two-hybrid assay was set up. We hypothesized that PIT4 is able to interact with a sensor kinase of a two-component system, likely FleS. However, we were surprised that PilS did not appear as an interaction partner in the Y2H, given its relevance in T4P function-ality. Therefore, two B2Hs were performed using both FleS and PilS as preys (Fig. 4). The bacterial two-hybrid assay confirmed the interaction between FleS and PIT4 when compared to the controls (Student's $t$-test, $P < 0.0001$). Interestingly, a positive result was also obtained for the interaction between PIT4 and PilS, although this interaction appeared weaker than the interaction between PIT4 and FleS (Student's $t$-test, $P < 0.0001$). This result can, in part, be explained by the intrinsic properties of the two proteins. PilS is a membrane-associated kinase and contains six hydrophobic regions. These features might imply that the PIT4-interacting region of PilS is less accessible and/or that PilS is incorrectly folded in the non-biological host cell. In contrast, FleS is

**TABLE 2** Identified protein-protein interactions by the Y2H screening

| | Identified prey protein | Corresponding *P. aeruginosa* gene (PAO1) | | | |
|---|---|---|---|---|---|
| | Locus tag | Gene | Length | Position | Confirmation one-on-one |
| 1 | PA1098 | fleS | 402 | 237–386 | No |
| | PA1098 | fleS | 402 | 164–386 | Yes |
| 2 | PA4197 | bfiS | 758 | 495–758 | Yes |
| 3 | PA2177 | Putative sensor/response regulator | 699 | 394–426 | Yes |
| 4 | PA2881 | Putative two-component regulator | 303 | 263–304 | No |
| | PA2882 | Putative two-component sensor | 371 | 1–235 | Yes |
| 5 | PA1243 | Probable sensor/response regulator | 858 | 562–758 | Yes |

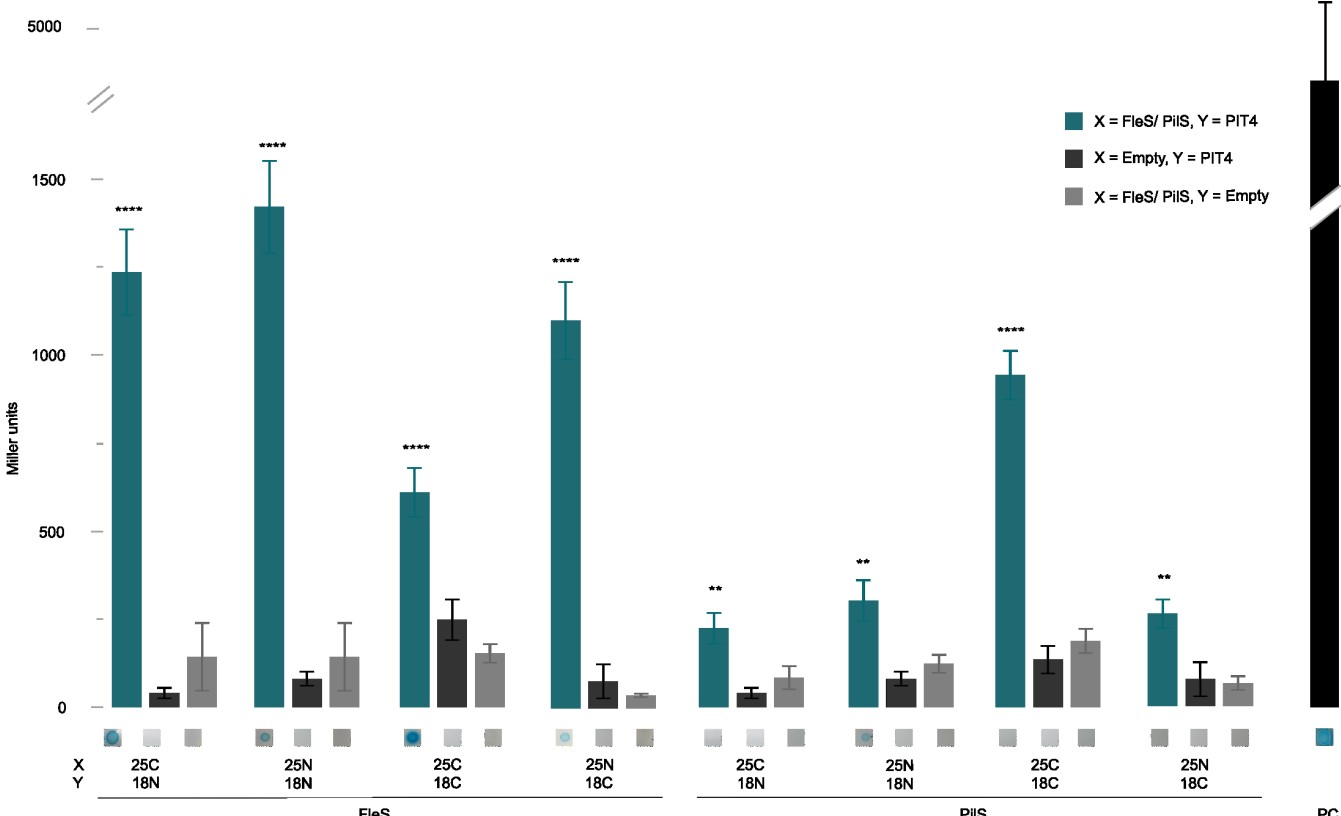

**FIG 4** Bacterial two-hybrid screening of PIT4 and PilS and FleS. In this interaction assay, a C-terminal or N-terminal fusion of the T25 or T18 domain of the adenylate cyclase CyaA to the protein of interest (X or Y) is made. As controls, non-fused T25 and T18 domains were used. the interactions were determined visually by a drop test on M63 selective medium containing α-X-galactosidase and quantitively by the determination of Miller units. An interaction of PIT4 with both FleS and PilS was observed (Student's *t*-test, **$P < 0.01$, ****$P < 0.0001$). The leucine zipper of GCN4 fused to the T18 and T25 domains served as a positive control.

present in the cytoplasm and is in this way more accessible for interaction with PIT4. It is therefore possible that under normal conditions, in the PAO1 cell, the interaction between PilS and PIT4 is stronger than captured by this B2H assay in *E. coli*. The obtained results suggest that PIT4 is able to interact with multiple histidine kinases of different TCSs, including FleS and PilS. To test this hypothesis, a second B2H experiment was performed with the histidine kinase PA2882 as prey. This bacterial protein was revealed in the aforementioned yeast two-hybrid screen and has a predicted HK function. Notably, this assay again resulted in a positive interaction between PA2882 and PIT4 (Fig. 5A; Student's *t*-test, $P < 0.0001$). Probably, the interaction region of the bacterial protein is a conserved region, for instance, present in the histidine kinase domain (percentage of identity: 28%–30%; Fig. S4). To confirm or reject this hypothesis, another bacterial two-hybrid experiment was performed to investigate if an interaction indeed occurred between PIT4 and the HK domain of FleS (Fig. 5B). This hypothesis was confirmed in both qualitative and quantitative ways (Student's *t*-test, $P < 0.0001$). In view of these findings, we can conclude that PIT4 interacts with at least three different histidine kinases of *P. aeruginosa* through interaction with the histidine kinase domains.

## PIT4 impacts the overall bacterial motility

Besides the aforementioned effect on the twitching motility, induction of *pit4* in *P. aeruginosa* leads to a inhibition of swimming and swarming motility compared to the control (Fig. 6A, Student's *t*-test, $P < 0.0001$; Fig. S5). The decreased cell motility cannot be attributed to a slower cell growth, as PIT4 production does not impact the growth rate

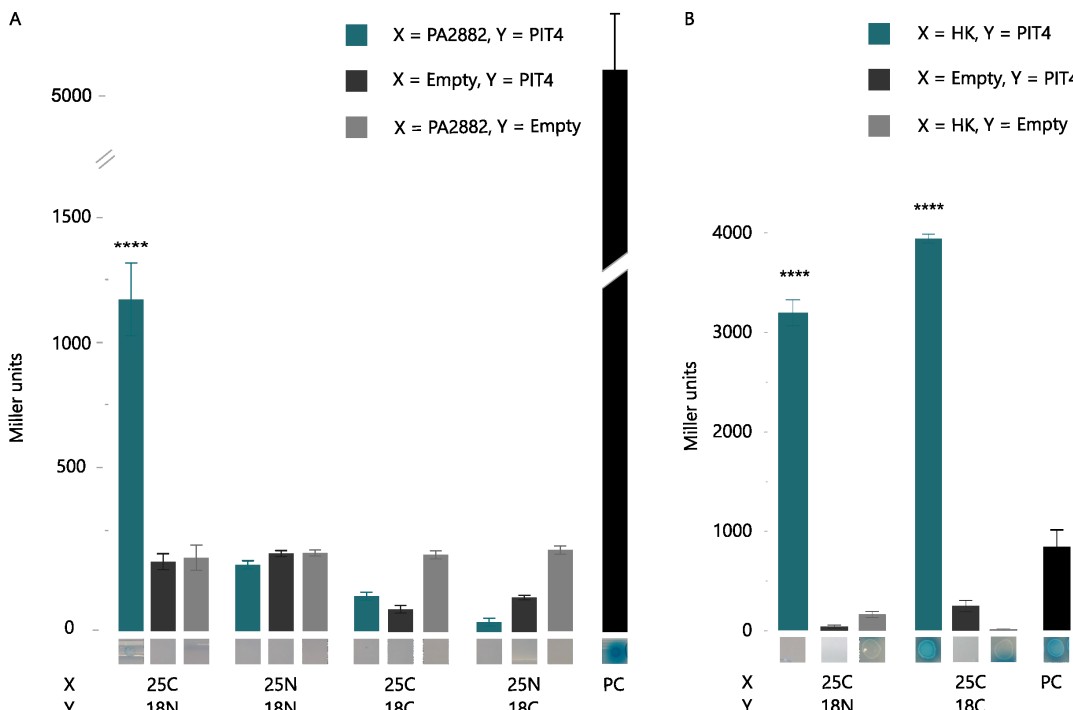

**FIG 5** Miller assay reveals the interaction between PIT4 and PA2882, as well as the HK domain of FleS. (A) In this bacterial two-hybrid assay, the gene encoding for the hypothetical histidine kinase PA2882 was fused C- and N-terminally to the T25 part of CyaA, and PIT4 was fused to the T18 part. The Miller assay revealed that an interaction took place between the two proteins (Student's *t*-test, ****$P < 0.0001$). (B) Interaction assays between PIT4 and the histidine kinase domain (HK) of FleS, the latter being fused C-terminally to the T25 part of CyaA. A strong interaction was observed between the HK domain and PIT4 (Student's *t*-test, ****$P < 0.0001$). The leucine zipper of GCN4 fused to the T18 and T25 domains served as a positive control.

of its host (Fig. 6B). Consistent with the effect that was observed on bacterial motility, expression of *pit4* also decreases biofilm formation (Fig. 6C; Student's *t*-test, $P < 0.01$). Taking together the observed results on twitching motility and biofilm formation, we can conclude that PIT4 has an overall effect on motility.

Because results of the performed bio-assays point to a change in the motility mechanism driven by PIT4, a phage infection assay was performed. Phage infection by phage LSL4 was followed over time (Fig. 6D). Production of PIT4 in the cell resulted in an observed inhibition of infection by phage LUZ19. Since LUZ19 uses the T4P as receptor, just like LSL4, the data further indicate an impairment of the T4P elicited by PIT4 (33, 34). As the number of known flagellotropic *Pseudomonas aeruginosa* phages is low, the effect of PIT4 on phage infection could not be determined for this type of phages (35). To further explore the identity of the pilus-like structures that we observed in TEM images, we used a fluorescent label specific to T4P pili and fluorescence microscopy to count T4P (36). Remarkably, this revealed the complete absence of T4P on the surface of the bacterial cell, upon expression of *pit4* (Fig. 7; Fig. S6). In addition, a transmission electron microscope (TEM) was used to visualize all the surface appendages. The observations made with TEM revealed that also the number of displayed flagella did significantly decrease upon *pit4* expression (Fig. S7; Table S2). This implies that the interaction of PIT4 with the HKs results in an aberrant number of flagella and T4P displayed on the bacterial surface, which might explain the effect observed on the different bacterial motilities. Moreover, only a limited number of pili-like structures on the cell surface of cells producing PIT4 were detected via TEM. However, as this technique is less accurate for the visualization of pili compared to the fluorescent microscopy experiment, we can assume that the pili-like structures that were seen with TEM are most likely not T4P. Taken together, the microscopy results indicate that both the flagellum and the T4P are not present on the bacterial cell if PIT4 is produced in the cell.

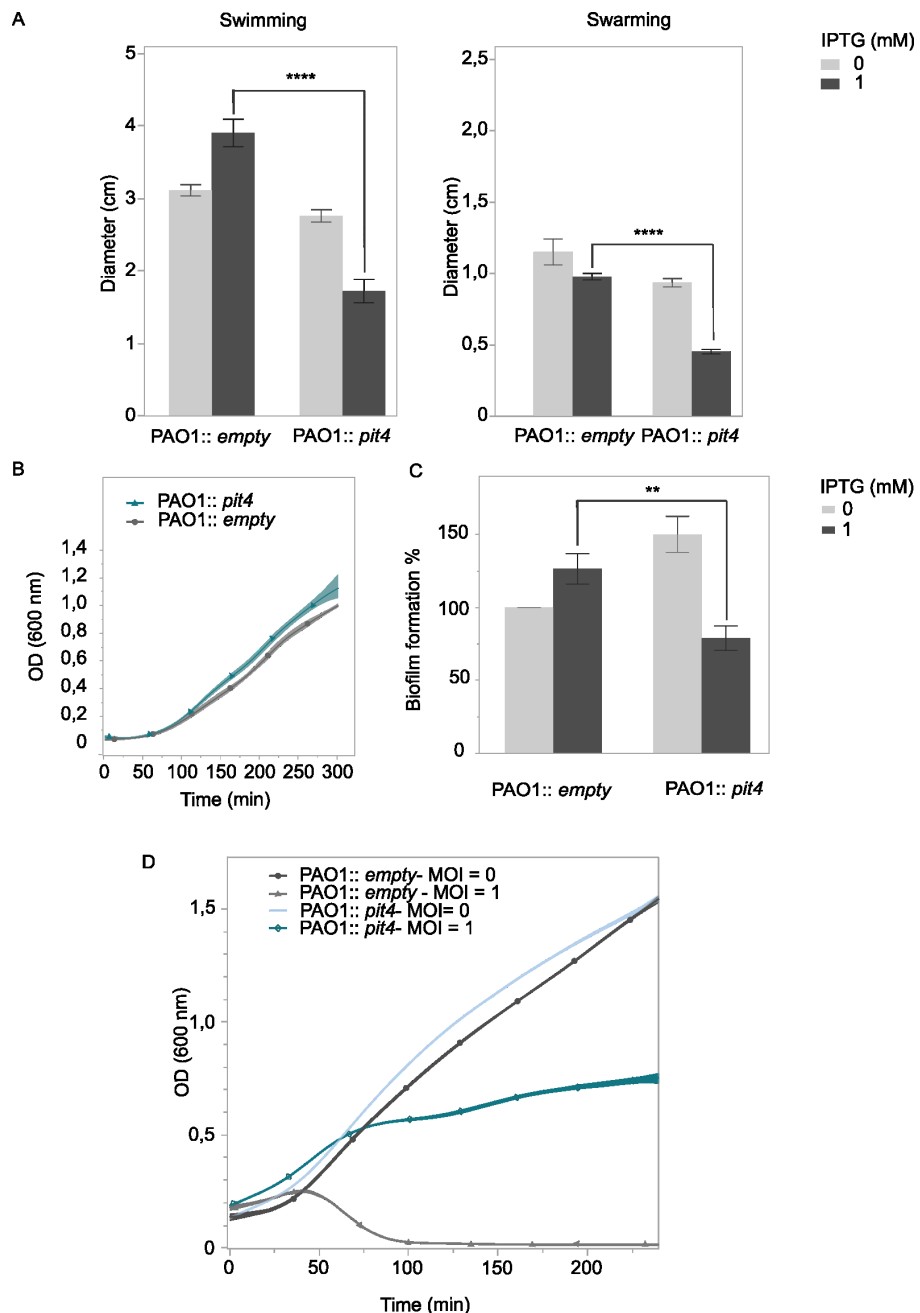

**FIG 6** PIT4 decreases bacterial motility, biofilm formation, and phage infection through the T4P. (A) Induction of PIT4 production in the PAO1 cell leads to a significant reduction of both swimming and swarming motility (Student's *t*-test, ****$P$ < 0.0001), when compared to the control. (B) The observed effects on motility are not resulting from a growth delay, as the growth of the PAO1 cells expressing pit4 is the same as for the control. (C) A reduced biofilm formation is observed for cells producing PIT4 (Student's *t*-test, **$P$ < 0.01). (D) Phage infection with LUZ19 results in a reduced ability of the phage to infect PAO1 expressing PIT4 through the T4P (PAO1::pit4) compared to the control (PAO1::empty).

## Conclusion

In this research, we identified the phage-derived protein PIT4 (phage inhibitor of the T4P) that interacts with multiple histidine kinases of different TCSs through the histidine kinase domain (Fig. 8). This interaction leads to the attenuation of bacterial motility and decreased functionality of the T4P. Future work will aim to elucidate the role of PIT4

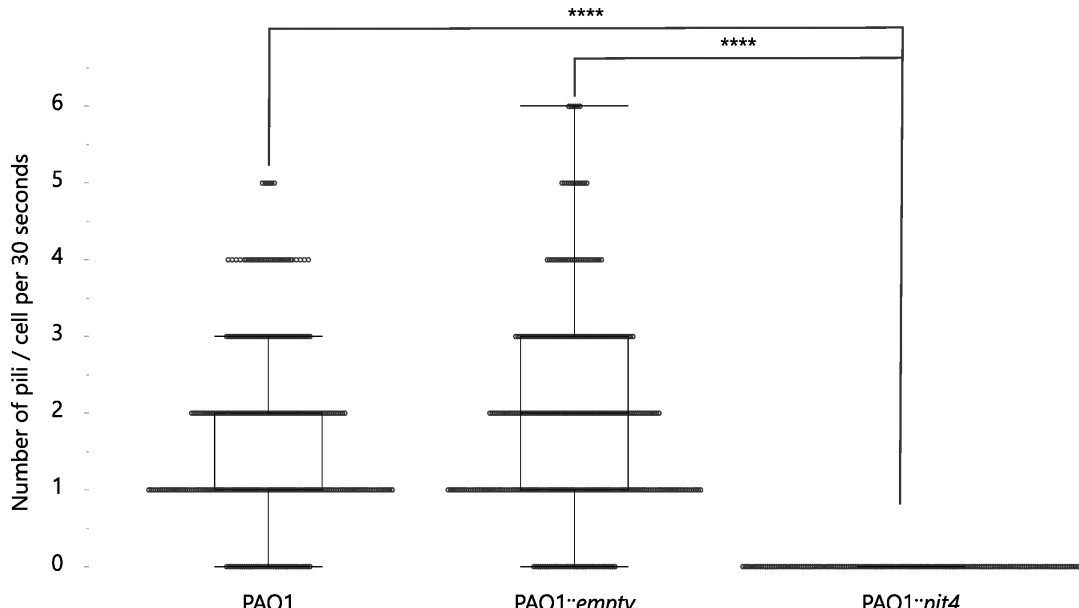

**FIG 7** Number of T4P observed by fluorescent microscopy. The number of T4P present on the bacterial surface was determined by making use of fluorescent microscopy. For each condition, three biological replicates of 50 cells each were analyzed for 30 s to determine the number of pili displayed. No pili structures were observed upon PIT4 production, in contrast to the PAO1 wild-type cells and PAO1 cells harboring an empty cassette (Student *t*-test, ****$P < 0.0001$).

during phage infection and identify the overall effects in the cell that can be contributed to these specific protein-protein interactions.

In *P. aeruginosa*, TCSs play a major role in the regulation of virulence-associated traits which allow the bacterium to promptly respond to an environmental stimulus (1). Surprisingly, cross-talk has been reported between the TCSs, suggesting that TCSs do not work on an individual level but rather are established in a multikinase network and in this way translate different environmental triggers into the desired cellular response (32, 37). PIT4 demonstrated to interact with several TCSs through interaction with the histidine kinase domain. This implies that, most likely, not all bacterial interaction partners were captured here and that additional TCSs will be affected by the phage-derived protein

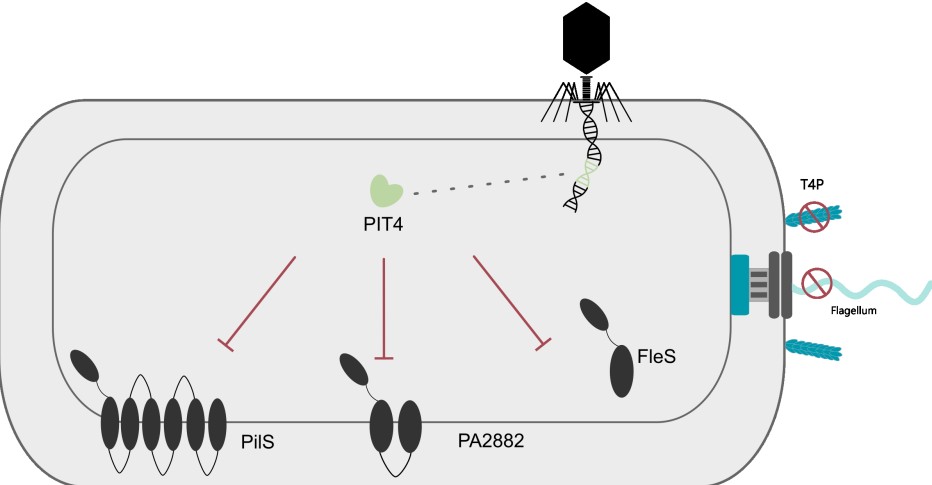

**FIG 8** Schematic overview of the proposed working mechanism of PIT4. When PIT4 is produced within the bacterial cell, it interacts with different histidine kinases of two-component systems. Via bacterial two-hybrid, the interaction with PilS, FleS, and PA2882 was confirmed. This interaction leads to different phenotypical changes, including the type IV pili (T4P) and flagellum.

as well. For instance, multiple TCSs, among others PhoQ/PhoP, ParS/ParR, and AmgS/ AmgR, have been shown to regulate the expression of antibiotic resistance genes upon the presence of antibiotics in the environment (38–40). Moreover, studies have shown that the AlgZ/AlgR system induces alginate production and twitching motility, while the BqsS/BqsR system modulates the biosynthesis of quorum sensing molecules and rhamnolipids (41, 42). At this point, only the effects of PIT4 on the motility moieties of the bacterial cell were captured. As such, it would be of interest to assess other virulence factors that are known to be regulated by TCSs to further quantify the impact of PIT4. From a biological perspective, the data presented hint toward a straightforward infection exclusion mechanism for phage LSL4. However, considering the widespread effect PIT4 has, additional functions cannot be excluded.

TCSs are widespread among all bacteria, including pathogens, and in many cases, the regulation of virulence traits is partly fulfilled by TCSs. For instance in *Salmonella enterica* and *Staphylococcus aureus*, a complex interplay between different TCSs contribute to the overall virulence (43). Moreover, due to their absence in mammals, these systems are a safe and suitable target (32). Interference with the signaling of HKs of TCSs has been recently postulated as an anti-virulence strategy against *P. aeruginosa* (44). The feature of PIT4 to interact and presumably interfere with different HKs enables the affection of key virulence factors with a single inhibitor. Indeed, through this interaction, PIT4 showed that it greatly affects the motility moieties, which play a crucial role in surface colonization and biofilm formation, putting it forward as an attractive anti-virulence target for a small molecule inhibitor (45).

## MATERIALS AND METHODS

### Bacterial and yeast strains, cells, plasmids, and growth conditions

In this study, four *Escherichia coli* strains were used: *E.coli* TOP10 (Thermo Fisher Scientific, Waltham, MA, USA) for the cloning experiments, *E. coli* BL21(DE3) (Thermo Fisher Scientific) for protein expression, *E. coli* KC8 (TaKaRa, Mountian, Shiga, Japan) for cloning procedures in the yeast two-hybrid experiment and *E. coli* BTH101 (Euromedex, Souffelweyersheim, France) for the bacterial two-hybrid assays. All the experiments in *P. aeruginosa* were performed in PAO1k (46). A transposon mutant of PilA (PW8622) was ordered from the *P. aeruginosa* mutant library (47). PAO1 strains harboring a single-copy genomic integration of a phage gene, under the expression of an isopropyl-β-D-1-thio-gamactopyranosie (IPTG)-inducible *lac* promoter and were created previously (17). The pBGDes plasmid was used for the expression of *pit4* under the control of the PEM7 constitutive promoter (48). The *Saccharomyces cerevisiae* AH109 strain was used in the yeast two-hybrid analysis (James et al., 1996 (49)). HeLa cells used in this research were a kind gift from Professor Johan Neyts, Laboratory of Virology and Chemotherapy (Rega Institute KU Leuven, Belgium).

The bacterial strains were grown in lysogenic broth (LB) at 37°C and supplemented, if necessary, with antibiotics unless stated differently. The antibiotics used in this study were ampicillin (Ap100, 100 µg/ mL), carbenicillin (Cb200, 200 µg/ mL), gentamycin (Gm30, 30 µg/ mL), kanamycin (Kan50, 50 µg/ mL), and tetracycline (Tc60, 60 µg/ mL). Yeast cells were grown in synthetic defined (SD) medium supplemented with the appropriate auxotrophic markers (50).

Transformations in *E. coli* were performed in chemically competent cells via the rubidium chloride method (51) followed by a heat shock (42°C, 30 s). Transformations in *P. aeruginosa* occurred via electroporation by using the Gene Pulser Xcell system (Bio-Rad, Hercules, CA, USA) 2.5 kV and 0.2-cm gap cuvettes (Bio-Rad) as described in the protocol of Choi et al. (52). Transformations into *S. cerevisiae* were performed in chemical competent cells according to the protocol of Wagemans et al. (50).

## Bacteriophages

Lytic phages LSL4, LUZ19, and LKA1 were used in this study and was amplified in PAO1 using the soft agar overlay method followed by PEG precipitation (53) and stored in phage buffer at 4°C (10-mM Tris-HCl, 10-mM MgS4, 150-mM NaCl; pH 7.5). For phage titer determination, a dilution series was spotted on top of a double agar plate (PFU/mL).

## Motility assays

To determine the motility of the PAO1 strains, different assays were performed. For the swimming and swarming assays, an overnight culture of the bacterial strains was grown and used to inoculate at the center of an agar plate with a toothpick (0.3% and 0.5% agar, respectively). Plates were inoculated overnight at either 37°C (swimming) or 30°C (swarming), followed by measurement of the spreading diameter. For the twitching assay, the protocol of Turnbull and Whitchurch was followed (54). Briefly, bacterial cells on an agar plate were collected, mixed, and picked with a toothpick, and the inoculum was stabbed at the agar-plastic interface of a 1% LB agar plate. The plates were incubated in a plastic container at 37°C overnight. The next day, the interstitial colony was determined, which corresponded to the twitching area. All assays were performed with five biological replicates.

## Biofilm assay

To assess the biofilm formation of the strains, the bacteria were inoculated in minimal M63 medium [15-mM $(NH_4)_2SO_4$, 100-mM $KH_2PO_4$, 1.7-µM $FeSO_4$, and 2% glycerol] supplemented with 0.4% L-arginine and 1-mM $MgSO_4$. The next morning, the cultures were diluted until an $OD_{600\,nm}$ of 0.02 was obtained. To the wells of a round bottom 96-well plate, 150 µL of the diluted bacterial cultures was added and 150 µL of M63 was added to the outer well, which served as blanks. The plate was sealed with a peg lid (Thermo Fisher Scientific) and closed with a single layer of parafilm and incubated for 24 hours at 37°C in a plastic box on an orbital shaker (125 g). After the incubation period, the peg lids were washed in phosphate-buffered saline (PBS) (137-mM NaCl, 2.7-mM KCl, 8.2-mM Na2HPO4, 1.8-mM KH2PO4; pH 7.4) and stained with crystal violet (2-mM crystal violet, 5% vol/vol methanol, 5% vol/vol isopropanol, and 90% vol/vol PBS) for 30 minutes. Next, the lids are washed three times in PBS, dried at room temperature, and destained in a 30% vol/vol acetic acid solution. The absorbance (600 nm) of the plate containing the acetic acid solution with dissolved crystal violet was measured in a microplate reader (CLARIOstar). The assay was performed with two technical replicates, each containing three biological replicates.

## Phage infection assay

To investigate the effect of PIT4 on phage infectivity, the PAO1::*empty* and PAO1::*pit4* strains were grown overnight and diluted the next morning in LB, supplemented with 1-mM IPTG for the induced conditions, and grown until the $OD_{600\,nm}$ reached 0.3. The bacterial cultures were infected with MOI of 0 or MOI of 1, and 200 µL of the infected culture was brought over into a 96-well plate. The $OD_{600\,nm}$ was measured for 6 hours, with 2-minute intervals. Three biological replicates were measured per condition.

## Differential gene expression

First, the RNA was extracted from the bacterial strains by diluting an overnight culture of PAO1::*empty* (control) or PAO1::*pit4* 100-fold in 50-mL LB (uninduced) or 50-mL/1-mM IPTG (uninduced). Four hours later, a total amount of $OD_{600\,nm} = 2$ cells were harvested; 20% ice-cold stop mix (95% vol/vol ethanol, 5% vol/vol phenol, saturated; pH 4.5) was added and immediately snap-frozen in liquid nitrogen. Subsequently, the cells were thawed on ice, the pellets were collected (20 min, 4°C, 4,500 $g$) and resuspended in 300-µL lysozyme solution [0.5 mg/mL in Tris-EDTA buffer (10-mM Tris-HCl, 1-mM EDTA;

pH = 8)]. RNA was extracted via a hot-phenol treatment and a subsequent ethanol precipitation. A DNase I treatment was performed to remove bacterial DNA, followed by a second ethanol purification. Genomic DNA removal was verified via a PCR reaction with a PAO1-specific primer pair. The quality of the resulting RNA was determined with the Bio-analyzer (Aligent 2100) using the RNA 6000 Pico kit. An RNA integrity number ≥9 was used as criterium for further downstream processing. The RNA was converted into cDNA with the Illumina Stranded Total RNA Prep Ligation Zero Plus Kit. The RNA depletion and average fragment length were evaluated by making use of the Bio-analyzer and the DNA high sensitivity kit (Aligent). The cDNA library was sequenced by Illumina next-generation sequencing (Novaseq 6000) in paired-end mode after pooling the samples in equimolar amounts until a total volume of 25 µL and 4 nM was obtained.

After sequencing, FastQC (v.0.11.8) (55) was used to evaluate sequencing yields and the quality of the raw reads, which were subsequently processed with Trimmomatic (v.0.29) (56) to remove poor quality bases and adapter sequences. The processed reads were aligned (BWA-MEM v.0.7.17) (57) to the reference sequence of *P. aeruginosa* PAO1 (NC_002516.2), which was manually modified to include the pit4 genomic insert in its Tn7 site. Next, the alignment files were converted with SAMtools (v.1.9) (58) to sorted and indexed BAM files that were visualized using Integrative Genomics Viewer (IGV) (v.2.8.9) (59). Next, assignment of the mapped read pairs (fragments) to the genomic features of PAO1::*pit4* was carried out using FeatureCounts (v.2.0.1) (60), followed by *in silico* removal of bacterial rRNA reads. Afterwards, the fragment counts were FPKM-normalized (fragment per kilobase of transcript per million mapped non-rRNA fragments) and subjected to principal component analysis in R (prcomp package) to assess sample clustering. Finally, the R Bioconductor package DESeq2 (v.1.30.0) (61) was used to perform differential gene expression analysis. Raw sequencing data and processed data files were deposited in the GEO database (GSE230851).

## Transmission electron microscopy

Transmission electron microscopy (TEM) was used to visualize the number of motility moieties displayed on the bacterial surface. First, the cells were fixed by centrifugation of overnight cultures of the bacterial strains (1,420 g, 15 min). Next, the supernatants were removed and the pellets were resuspended in 2.5% v/v glutaraldehyde in 100 mM phosphate buffer (50 mM $Na_2HPO_4$, 50 mM $NaH_2PO_4$; pH 7) and incubated overnight at 4°C while shaking. The next day, the sample were centrifuged (1420 g, 5 min, 4°C) and resuspended in 100 mM phosphate buffer. The bacterial cells were negatively stained with 0.5% w/v uranyl acetate in water. Images were obtained by using the Philips CM10 TEM. For each condition, forty biological replicates were assessed.

## Fluorescent microscopy

For the visualization of PilA under the fluorescent microscope, a point mutation was introduced *pilA*[A86C]. The mutation was brought into the PAO1 genome using the Cas3cRh and pSEVA131 plasmids via CRISPR-Cas engineering (62, 63). Next, a pUT18-mini-Tn7T plasmid harboring the *pit4* gene and an empty cassette were introduced separately into the PAO1(*pilA*[A86C]) strain.

Imaging of T4P was performed as described previously (36). Briefly, overnight cultures were diluted 1:500 and grown with or without 1-mM IPTG as inducer for 3 hours in LB medium at 37°C shaking. Of 2.5 mg/mL AlexaFluor 488 Cysteine maleimide dye (Fisher), 1.8 µL was added to 180 µL of cells and incubated for 45 min. Cells were washed twice by pelleting using centrifugation and resuspension in EZ rich medium. 1 µL of labeled cell suspension was added to a small 1% agarose pad and transferred on a glass bottom petri dish for imaging. Images were taken on an inverted Nikon Ti2 microscope with ×100 NA 1.45 lens and a Hamamatsu FusionBT camera. Videos were taken at a frame rate of 2 Hz for 30 s. The experiment was performed by analyzing 50 biological repeats per condition.

## Assessment of the virulence in HeLa cells

The viability of HeLa cells was measured based on the protocol described in reference (64). Overnight PAO1 cultures were diluted and grown until $OD_{600\,nm}$ of 0.3 followed by centrifugation (4,600 $g$, 4 min) and resuspension of the pellets in Dulbecco's modified eagle medium (DMEM), supplemented with 10% vol/vol FBS and 1% vol/vol minimal essential medium non-essential amino acids (MEM NEAA). The bacterial suspension was diluted until $10^{-5}$ and 10 µL of the suspension was added to the human cells. These cells were cultured in DMEM supplemented with 10% vol/vol FBS, 1% vol/vol antibiotic-anti-mycotic solution, 1% vol/vol MEM NEAA and 1% vol/vol sodium bicarbonate and incubated at 37°C (5% $CO_2$). Once the cells were confluent, they were seeded in a 96-well plate and incubated overnight (37°C, 5% $CO_2$). After an incubation period of 16 hours, bacterial cells were removed from the microtiter plate and washed twice with sterile PBS, and the viability of the cells was measured using the MTT cell viability kit (Sigma-Aldrich) and read out in the CLARIOstar plus microplate reader. In addition, the viability of the exposed HeLa cells was analyzed under the microscope (EVOS FL Auto imaging system, Thermo Fisher Scientific). The experiment was performed with six biological replicates for each strain.

## *In vivo G. mellonella* experiment

The *P. aeruginosa* strains were grown overnight and diluted the next day, and when they reached an OD of 0.3, the cells were pelleted (4,600 $g$, 4 min). The pellets were resuspended in sterile PBS and diluted until $10^{-5}$. Next, 10 µL of the dilution, corresponding to approximately 10 cells, was injected into the hindmost proleg of the larvae with a thin needle (Microfine[+] insulin). The larvae were placed in a sterile petri dish and incubated in the dark at 37°C for 24 hours, after which their survival rate was assessed. Eight larvae were used to assess each bacterial strain.

## Yeast two-hybrid

To identify putative interactions between PIT4 and PAO1 proteins, a Y2H assay was performed according to the protocol of Wagemans and Lavigne (50). *Pit4* was fused to the Gal4p DNA-binding domain in the pBGT9 plasmid and introduced into chemical competent AH109 *S. cerevisiae* cells. Next, a previously obtained prey library of random genomic PAO1 fragments (13) was transformed into the bait-containing AH109 cells. Selection of positive clones was done on SD minimal medium complemented with 5-bromo-4-chloro-3-indolyl-α-D-galactopyranoside.

## Bacterial two-hybrid

The bacterial two-hybrid experiment was performed with the BATCH system (Bacterial Adenylate Cyclase Two-Hybrid system kit, Euromedex). As preys, the *P. aeruginosa* genes *fleS*, *pilS* and *PA2882* were fused C- and N-terminally to the T25 unit, and *pit4* was fused to the T18 subunit C- and N-terminally and served as bait. For the assessment of the HK domain of FleS, a site-directed mutagenesis was performed by making use of a specific primer pair to only amplify the pKT25 plasmid and the HK domain. Each prey-bait combination was co-transformed into competent *E.coli* BTH101 cells and spotted on both LB and minimal M63 medium [15-mM $(NH_4)_2SO_4$, 100-mM $KH_2PO_4$, 1.7-µM $FeSO_4$, 1-mM $MgSO_4$, 0.05% (wt/vol) vitamin B1, 20% (wt/vol) maltose, 1.5% (wt/vol) agar] supplemented with Ap100, Kan50, 1-mM IPTG, and 40-µg/ mL 5-bromo-4-chloro-3-indolyl-β-D-galactopyranoside (X-gal) and incubated for 2–7 days at 30°C. The Miller assay was performed for quantification of the interaction. Each combination was tested in triplicate, and empty vectors were co-transformed with their counterparts and were used as negative controls. As a positive control, the pKT25(zip) and pUT18C(zip) plasmids were co-transformed to the *E. coli* cells.

## Statistical analysis

The information about the performed statistical test as well as the standard deviations and significance levels (*P* values) can be found in the figure legends or throughout the text. All calculations were done with the statistical software JMP (JMP Pro v.16).

## ACKNOWLEDGMENTS

We thank Prof. Abram Aertsen (Department of Microbial and Molecular Systems, KU Leuven, Belgium) for the provided suggestions and insights and the former MSc thesis student Lizze Boonen for the conduction of the Y2H experiment.

K.S. and R.L. conceived and designed the research and wrote the manuscript. K.S. and A.K. conducted the experiments. P.J.C. isolated and sequenced phage LSL4. L.P. performed the data analysis of the differential gene expression experiment. M.V. carried out the transmission electron microscope experiment. J.P. provided instruments and insight about the HeLa experiment. F.H., A.Y., and M.D.K. designed, performed, and analyzed T4P fluorescence experiments. All authors read and approved the manuscript.

## AUTHOR AFFILIATIONS

[1]Laboratory of Gene Technology, KU Leuven, Leuven, Belgium
[2]Institute of Sustainable Plant Protection, National Research Council of Italy, Turin, Italy
[3]Host and Pathogen Interactions, KU Leuven, Leuven, Belgium
[4]Department of Biology, Texas A&M University, College Station, Texas, USA

## PRESENT ADDRESS

Pieter-Jan Ceyssens, Bacterial Diseases, Sciensano, Brussels, Belgium

## AUTHOR ORCIDs

Kaat Schroven  http://orcid.org/0000-0002-3663-8343
Leena Putzeys  http://orcid.org/0000-0002-5363-7764
Jan Paeshuyse  http://orcid.org/0000-0003-2396-354X
Rob Lavigne  http://orcid.org/0000-0001-7377-1314

## FUNDING

| Funder | Grant(s) | Author(s) |
| --- | --- | --- |
| EC | European Research Council (ERC) | 819800 | Rob Lavigne |

## AUTHOR CONTRIBUTIONS

Kaat Schroven, Conceptualization, Investigation, Methodology, Validation, Visualization, Writing – original draft | Leena Putzeys, Investigation | Alison Kerremans, Investigation | Pieter-Jan Ceyssens, Writing – review and editing | Marta Vallino, Investigation | Jan Paeshuyse, Resources, Writing – review and editing | Farhana Haque, Investigation | Ahmed Yusuf, Investigation | Matthias D. Koch, Investigation, Writing – review and editing | Rob Lavigne, Conceptualization, Funding acquisition, Resources, Supervision, Writing – review and editing

## DATA AVAILABILITY

The genome of phage LSL4 was deposited in NCBI GenBank (GenBank accession number OQ970155). Raw RNA sequencing files were deposited under GEO accession number GSE230851. Any additional information is accessible from the authors upon request.

## ADDITIONAL FILES

The following material is available online.

## Supplemental Material

**Supplemental material (Spectrum02372-23-s0001.docx).** Fig. S1 to S7; Tables S1 to S4.

## Open Peer Review

**PEER REVIEW HISTORY (review-history.pdf).** An accounting of the reviewer comments and feedback.

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
