## [Reviewer comments · Microbiology Spectrum]

Microbiology Spectrum

The phage-encoded PIT4 protein affects multiple two-component systems of *Pseudomonas aeruginosa*

Kaat Schroven, Leena Putzeys, Alison Kerremans, Pieter-Jan Ceysens, Marta Vallino, Jan Paeshuyse, Farhana Haque, Ahmed Yusuf, Matthias Koch, and Rob Lavigne

Corresponding Author(s): Rob Lavigne, Katholieke Universiteit Leuven

Review Timeline:

Submission Date:	June 8, 2023
Editorial Decision:	August 11, 2023
Revision Received:	September 25, 2023
Accepted:	October 4, 2023

Editor: Daria Van Tyne

Reviewer(s): Disclosure of reviewer identity is with reference to reviewer comments included in decision letter(s). The following individuals involved in review of your submission have agreed to reveal their identity: Shuguang Lu (Reviewer #2)

Transaction Report:

DOI: <https://doi.org/10.1128/spectrum.02372-23>

August 11, 2023

Dr. Rob Lavigne
Katholieke Universiteit Leuven
Leuven
Belgium

Re: Spectrum02372-23 (The phage-encoded PIT4 protein affects multiple two-component systems of *Pseudomonas aeruginosa*)

Dear Dr. Rob Lavigne:

Thank you for submitting your manuscript to Microbiology Spectrum. Your manuscript was reviewed by two experts and I would now like you to revise your study in line with their feedback below.

Link Not Available

Sincerely,

Daria Van Tyne
Editor, Microbiology Spectrum
Journals Department
Reviewer comments:

Reviewer #1 (Comments for the Author):

This MS reports the identification of phage LSL4 protein PIT4 and characterization of its influence on *P. aeruginosa* motility, gene expression, and phage infection. The authors show that single copy expression of PIT4 significantly reduces twitching, swarming motility, and swimming motility. To better understand how PIT4 reduces motility, the authors assessed PIT4's influence on *P. aeruginosa* gene expression by RNA sequencing. This revealed that genes encoding flagellar biosynthesis, a few type 4 pili genes, and genes involved in c-di-GMP metabolism were down regulated compared to the control. To elucidate the mechanism by which PIT4 acts, the authors determined PIT4 protein interactions with a yeast two-hybrid and bacterial two-hybrid experiments. They show that PIT4 interacts with a few histidine kinases, specifically PilS (T4P), FleS (flagella), and the histidine kinase domain of FleS. This suggests that PIT4 exerts control of *P. aeruginosa* motility by interaction with TCS histidine kinase domains. Of note, the phage that encoded PIT4, phage LSL4, uses T4P as a receptor for adsorption. The authors demonstrate that overexpression of PIT4 reduces phage infection compared to the control. Interestingly, they show that the

number of pili on the bacterial surface is not effected by PIT4 expression, suggesting that PIT4 influences T4P function which subsequently reduces twitching motility and phage infection. Interestingly, the authors demonstrate that overexpression of PIT4 reduces the number of flagella on the bacterial surface, but does not significant impact phage infection by a phage that uses flagella as a receptor. The manuscript is well written and logically organized and I believe the data reported would be of interest to the readers of Microbiology Spectrum. I have a few minor comments for the authors consideration.

Minor Comments:

Line 166-169 : You indicate that *gcbA* and *siaD* play a role in c-di-GMP levels. I think the readers would benefit from additional discussion of how these results are consistent with the known roles of these proteins and the observed motility phenotype

Line 254-257: The readers may benefit by additional discussion of why the inactivation of T4P would impair phage infection, but that reduction of flagella and flagellar activity did not prevent flagellotropic phage infection.

Reviewer #2 (Comments for the Author):

In this study, a phage derived protein was studied that impacts key virulence factors of the pathogen via interaction with multiple TCSs. The fundamental insights gained for this protein can therefore serve as inspiration for the development of an anti-virulence compounds that target the bacterial TCSs.

Main concerns:

Figure 1: The strains in Figure 1 (PAO1 with different overexpression plasmids) exhibited consistent phenotypes in twitching without IPTG addition. However, the strains in the figure 6 (PAO1 with different overexpression plasmids) exhibited inconsistent motility phenotypes without the addition of IPTG. Why? Even though this difference is caused by the expression of the plasmid itself, some differences are indeed quite significant.

The full text mentions a two component system, but only studies the interaction between one gene in the system, without specifying whether there is an interaction with the other gene.

The phenotype experiment only saw statistical data and did not see photo images, especially the two component system that affects phenotype changes. There are no specific diagrams showing biofilms, swimming, swarming, twitching.

The topic is mechanism research, but I did not see a clear explanation of the mechanism, and I feel that the explanation is not clear.

Figure 4: Should a positive control group be set up for bacterial double hybridization experiments? (Positive : pUT18C-zip+pKT25-zip)

The results of protein interaction should be further verified by Pull down or other methods. Should Transcriptome data also be verified?

The language needs to be more rigorous. Such as Line 46 (TCSs), Line 89 (in the regulation of phage), Line 92 (lytic P. aeruginosa phage LSL4), Line 332 (PFU/mL), etc.

Staff Comments:

Preparing Revision Guidelines

Please return the manuscript within 60 days; if you cannot complete the modification within this time period, please contact me. If you do not wish to modify the manuscript and prefer to submit it to another journal, please notify me of your decision immediately so that the manuscript may be formally withdrawn from consideration by Microbiology Spectrum.

Reviewer comments:

Reviewer #1 (Comments for the Author):

This MS reports the identification of phage LSL4 protein PIT4 and characterization of its influence on *P. aeruginosa* motility, gene expression, and phage infection. The authors show that single copy expression of PIT4 significantly reduces twitching, swarming motility, and swimming motility. To better understand how PIT4 reduces motility, the authors assessed PIT4's influence on *P. aeruginosa* gene expression by RNA sequencing. This revealed that genes encoding flagellar biosynthesis, a few type 4 pili genes, and genes involved in c-di-GMP metabolism were down regulated compared to the control. To elucidate the mechanism by which PIT4 acts, the authors determined PIT4 protein interactions with a yeast two-hybrid and bacterial two-hybrid experiments. They show that PIT4 interacts with a few histidine kinases, specifically PilS (T4P), FleS (flagella), and the histidine kinase domain of FleS. This suggests that PIT4 exerts control of *P. aeruginosa* motility by interaction with TCS histidine kinase domains. Of note, the phage that encoded PIT4, phage LSL4, uses T4P as a receptor for adsorption. The authors demonstrate that overexpression of PIT4 reduces phage infection compared to the control. Interestingly, they show that the number of pili on the bacterial surface is not effected by PIT4 expression, suggesting that PIT4 influences T4P function which subsequently reduces twitching motility and phage infection. Interestingly, the authors demonstrate that overexpression of PIT4 reduces the number of flagella on the bacterial surface, but does not significant impact phage infection by a phage that uses flagella as a receptor. The manuscript is well written and logically organized and I believe the data reported would be of interest to the readers of Microbiology Spectrum. I have a few minor comments for the authors consideration.

We would like to thank the reviewer for these kind words and the recognition of the relevance of the study within the field.

Minor Comments:

Line 166-169 : You indicate that *gcbA* and *siaD* play a role in c-di-GMP levels. I think the readers would benefit from additional discussion of how these results are consistent with the known roles of these proteins and the observed motility phenotype

We agree with the reviewer that this would be an asset for the paper. In a recent paper (Zhou, 2021), they found that FleS/R is regulating the expression of several genes, including GcbA and siaD. We added this explanation to the main text.

Line 254-257: The readers may benefit by additional discussion of why the inactivation of T4P would impair phage infection, but that reduction of flagella and flagellar activity did not prevent flagellotropic phage infection.

After receiving this comment, we looked deeper into the phages we used in this research. We found out that LKA1, the phage that was used in this work, can infect through the LPS. However, almost no flagellotropic Pseudomonas aeruginosa phages are described and we do not have a strictly (known) flagella-dependent phage at our lab. We can therefore not include a correct infection curve for this type of phage in the manuscript, nor can we argue that phage infection is hampered/ not affected. Nonetheless, based on the TEM results, we can conclude that the flagellum is absent upon PIT4 expression. Therefore, we adapted this in the manuscript. We would like to thank the reviewer for this comment, as this revealed our mistake.

Reviewer #2 (Comments for the Author):

In this study, a phage derived protein was studied that impacts key virulence factors of the pathogen via interaction with multiple TCSs. The fundamental insights gained for this protein can therefore serve as inspiration for the development of an anti-virulence compounds that target the bacterial TCSs.

We thank the reviewer for her/his positive feedback and useful comments.

Main concerns:

Figure 1: The strains in Figure 1 (PAO1 with different overexpression plasmids) exhibited consistent phenotypes in twitching without IPTG addition. However, the strains in the figure 6 (PAO1 with different overexpression plasmids) exhibited inconsistent motility phenotypes without the addition of IPTG. Why? Even though this difference is caused by the expression of the plasmid itself, some differences are indeed quite significant.

Figure 1 shows that when IPTG is added, PAO1 has a higher twitching capacity (as illustrated by the empty strain PAO1::empty). However, a reduction in twitching is noticed when different phage genes are expressed (LUZ19gp25.1 and LSL4gp67). In Figure 6, the same phenomenon is observed, namely that IPTG is increasing the twitching motility of the empty strain (panel C). The same observation was made for swimming motility (panel A). However, this increase is not seen for swarming motility, where a decrease, yet not significant, in motility is observed for the strains harboring an empty cassette.

The full text mentions a two component system, but only studies the interaction between one gene in the system, without specifying whether there is an interaction with the other gene.

We do apologize for this and thank the reviewer for bringing this up. We made it more clear with which component PIT4 is interacting.

The phenotype experiment only saw statistical data and did not see photo images, especially the two component system that affects phenotype changes. There are no specific diagrams showing biofilms, swimming, swarming, twitching.

We added the pictures of the twitching, swimming and swarming assays in the supplementary files (Fig S5). Moreover, data was added of a fluorescent microscopy experiment, showing the absence of T4P upon PIT4 expression (Figure 7 and Figure S6).

The topic is mechanism research, but I did not see a clear explanation of the mechanism, and I feel that the explanation is not clear.

We apologize for this unclarity. To make sure the mechanism behind this phage protein is understood by the reader, we added a figure to the manuscript that summarizes our findings and might help with understanding the mechanism behind the phage protein. On the other hand, we discussed the biological relevance of the protein in our manuscript, which is the main focus here.

Figure 4: Should a positive control group be set up for bacterial double hybridization experiments? (Positive : pUT18C-zip+pKT25-zip)

A positive control was included in the experiments but not displayed in our initial figures, as these values are many times higher than those tested. For the convenience of the reader, we did now include these positive controls in both figures (Fig 5 & Fig 6).

The results of protein interaction should be further verified by Pull down or other methods. Should Transcriptome data also be verified?

A pull-down experiment was performed by making use of a PAO1 lysate and a Strep-tagged PIT4. However, the interactions that were encountered in the previous assays (B2H and Y2H) could not be confirmed via this method, presumably due to the reason that these proteins have transmembrane components. Below you can find the table with proteins that were pulled down together with PIT4 and analyzed via Mass Spectrometry. One could notice that the amount of PIT4 retrieved is relatively low in the combinatory sample (PAO1 lysate + PIT4). Apart from the PilA molecules that were observed, no other links to the data that was collected previously (bioassays, differential RNA sequencing, Y2H and B2H).

Protein	PAO1 lysate + PIT4	PIT4	PAO1 lysate
PIT4	5	41	0
accB	85	18	13
accB	17	5	0
groL	13	0	3
liuB	5	0	1

atpF	5	0	0
rpsB	10	0	0
rplE	2	0	0
pilA	8	0	0
PA0070	5	0	0
PA5258	1	0	0
rpsE	6	0	0
rpsK	3	0	0
rplX	1	0	0
rplU	1	0	0
rplT	1	0	0
hupB	3	0	0
gcvP	1	0	0
grxC	1	0	0
rpsP	3	0	0
rplQ	2	0	0
PA3468	1	0	0

The language needs to be more rigorous. Such as Line 46 (TCSs), Line 89 (in the regulation of phage), Line 92 (lytic *P. aeruginosa* phage LSL4), Line 332 (PFU/mL), etc.

Thank you for raising this. We made the suggested adaptations in the text.

October 4, 2023

Dr. Rob Lavigne
Katholieke Universiteit Leuven
Leuven
Belgium

Re: Spectrum02372-23R1 (The phage-encoded PIT4 protein affects multiple two-component systems of *Pseudomonas aeruginosa*)

Dear Dr. Rob Lavigne:

Your manuscript has been accepted, and I am forwarding it to the ASM Journals Department for publication. You will be notified when your proofs are ready to be viewed.

Sincerely,

Daria Van Tyne
Editor, Microbiology Spectrum
